# An ENAS Based Approach for Constructing Deep Learning Models for Breast Cancer Recognition from Ultrasound Images

**Mohammed Ahmed**[*1]                                   1200526@buckingham.ac.uk
**Hongbo Du** [1]                                        hongbo.du@buckingham.ac.uk
**Alaa AlZoubi** [1]                                     alaa.alzoubi@buckingham.ac.uk
[1] *School of Computing, University of Buckingham, UK*

## Abstract

Deep Convolutional Neural Networks (CNN) provides an "end-to-end" solution for image pattern recognition with impressive performance in many areas of application including medical imaging. Most CNN models of high performance use hand-crafted network architectures that require expertise in CNNs to utilise their potentials. In this paper, we applied the Efficient Neural Architecture Search (ENAS) method to find optimal CNN architectures for classifying breast lesions from ultrasound (US) images. Our empirical study with a dataset of 524 US images shows that the optimal models generated by using ENAS achieve an average accuracy of 89.3%, surpassing other hand-crafted alternatives. Furthermore, the models are simpler in complexity and more efficient. Our study demonstrates that the ENAS approach to CNN model design is a promising direction for classifying ultrasound images of breast lesions.

**Keywords:** Efficient Neural Architecture Search, Ultrasound Image, Breast Cancer, Deep Learning

## 1. Introduction

Breast Cancer is one of the most common and life-threatening cancers in the world (DeSantis et al., 2019). Early diagnosis of breast cancer enables timely treatment, increasing the rate of survival for patients (Ponraj et al., 2011). Computer-Aided Diagnosis systems have been developed for classifying US images of breast lesions (Erickson et al., 2017). Deep learning is considered a significant technology breakthrough as it has exhibited good performance in image object classification for various applications, and many attempts have been made in using CNN for cancer recognition from US images (Litjens et al., 2017). However, no CNNs have been designed and optimized automatically for medical US image classifications, and most research efforts focus on adapting existing CNN architectures with or without transfer learning (Altaf et al., 2019). Adapting an existing CNNs of many hyperparameters requires expertise and makes the resulting models complex with increased risk for model overfitting (Elsken et al., 2018). Neural Architecture Search (NAS) is a recent development in optimizing CNN architectures (Zoph and Le, 2016).Although the method outperforms the hand-crafted CNN architectures for natural image classification, the optimisation process is

---

[*] Contributed equally

time consuming and the resulting models still have complex structures (Elsken et al., 2018). The follow-up work, known as Efficient Neural Architecture Search (ENAS), searches for optimal CNN architectures by using an RNN controller with reinforcement learning (RL) capability with less hardware requirements (Pham et al., 2018). Most recently, ENAS was applied for medical image segmentation such as (Gessert and Schlaefer, 2019) and (Weng et al., 2019).

In this study, we apply the ENAS approach to search for an optimal CNN architecture for breast cancer classification from US images. Due to its appeal of simplicity, the micro search space is adopted to generate optimal cells based on US images of breast lesions. The optimal cells are then selected for constructing the entire CNN architecture, followed by model training and testing.

## 2. Data and Methods

### 2.1. Data Pre-processing

A dataset of 524 US images of breast lesions (262 benign and 262 malignant) were collected from Shanghai Pudong New District People's Hospital. The lesion status (benign or malignant) for each image was confirmed by histopathological assessment of tissue samples and served as the ground-truth for this study. Besides, the boundaries of each lesion, i.e. Region of Interest (RoI), were identified and cropped manually by an experienced radiologist from the hospital. Figure 1 illustrates two example RoI images cropped from the original US images. To enlarge the training dataset, the following data augmentation methods have been applied on each RoI image (not the original US image): geometric methods (mirroring and rotation with degrees 90, 180 and 270 respectively) and singular value decomposition (SVD) with 45%, 35% and 25% ratios of the selected top singular values. These data augmentation methods generated 7 extra images from each RoI image. The input image size is rescaled to 100×100 using bicubic interpolation.

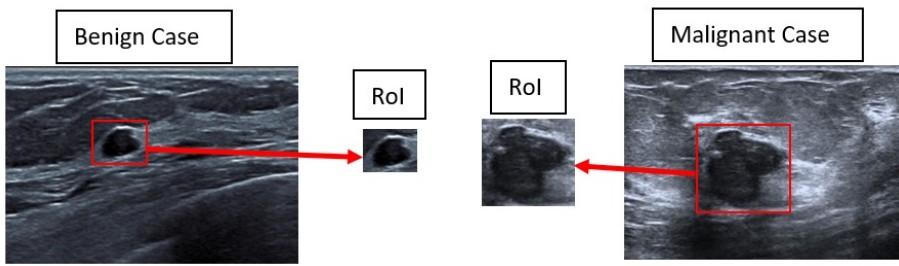

Figure 1: Examples of ultrasound images with labeled region of interest

### 2.2. Searching for CNN Architecture Using ENAS

The first stage of ENAS is to search for optimal cells (Normal (N) and Reduction (R)). The operations provided for the controller to generate cells are identity, separable convolution

with kernel sizes 3 and 5, average pooling and max pooling with kernel size 3. All hyper-parameters are set to the default values of ENAS. The number of epochs for the controller training is set to 150. At each epoch, the controller generates 10 candidate architectures, which are then evaluated using a validation set that is 10% of the training set. Based on the validation accuracy, the optimal cells are selected in a structure as shown in Appendix A. The second stage is to construct the optimized CNN by stacking the optimal cells from the first stage. For this study, we created two optimized CNNs: ENAS 7 (7 layers(N, R, N, R, N×3)) and ENAS 17 (17 layers (N×5, R, N×5, R, N×5)). Once the CNNs are built, the training set is used to train the network-based classification models from scratch through 100 epochs. For this study, 5-fold stratified cross validation was used for evaluating model performances.

## 3. Experiment Results and Discussion

Table 1 summarizes the performance metrics of the optimized ENAS 7 and ENAS 17 archi-tectures on the given data set of US images. ENAS 17 shows all round good performance with an overall accuracy of 89.3% and True Positive Rate (TPR) of 92%. For ENAS 7, the overall accuracy is marginally reduced because of a near 6% reduction in TPR but near 4% and 3% increase in True Negative Rate (TNR) and Precision (PR), respectively. At the expense of the marginal reduction of accuracy, the complexity of the ENAS 7 architecture is reduced by nearly 50% comparing to ENAS 17.

For comparison, we selected two alternative hand-crafted CNNs: the basic AlexNet (Krizhevsky et al., 2012) and CNN3 (Xiao et al., 2018). AlexNet (Krizhevsky et al., 2012) is one of the commonly use CNN architecture that is originally designed for natural image classification. CNN3 (Xiao et al., 2018) is a more recently designed architecture for breast lesion classification from US images. For fair comparison on network architectures, we purposely excluded any use of transfer learning, and all models were trained from scratch in the same folds of RoI images and evaluated with the same folds of RoI images. The hyperparameters were fixed as the default. The only alternation made to the AlexNet is the number of nodes in the last fully connected layer, which is reduced to two nodes according to the number of classes (Benign and Malignant). The performance of the models on the basic AlexNet is close to random guesses, and the network has 13 times as many weight parameters as ENAS 17 and 24 times as many as ENAS 7. CNN3 does have the lowest model complexity, but CNN3-based models have poorer performances than those on the two ENAS optimal architectures.

Table 1: ENAS Model Performance and Comparison with Other Architectures Models

| Models | TNR | TPR | PR | Accuracy | No. Parameters |
|--------|-----|-----|-----|----------|----------------|
| ENAS 17 | 86.7% | 92.0% | 87.5% | **89.3%** | 4251780 |
| ENAS 7 | 90.9% | 86.7% | 91.0% | 88.8% | 2342484 |
| AlexNet | 51.6% | 48.5% | 50.0% | 50.0% | 56858656 |
| CNN3 | 80.5% | 75.6% | 84.0% | 78.1% | 619202 |

## 4. Conclusion

This work investigated the efficacy of the ENAS approach for designing CNN architectures for breast cancer classification from ultrasound images. This study demonstrates that the ENAS technique reduces human interventions in CNN architecture design, and the optimised architectures lead to more accurate classification models than hand-crafted alternatives for breast cancer classification. Intrigued by the results, we plan to investigate further the effectiveness of ENAS architectures (particularly generic architectures) for different types of cancer from ultrasound images. We also plan to extend our work by comparing the performance of ENAS against the recent state of the art CNN architectures such as VGG, Inceptions, ResNet and MobileNet for breast US image classification. Finally, we wish to investigate the effect of the geometrical augmentation method (rotation) that transform the shape of the tumour on the classification performance of lesions.

## Acknowledgments

This research is sponsored by TenD Innovations. The authors wish to thank Dr Yicheng Zhu of Department of Ultrasound, Shanghai Pudong New District People's Hospital for providing the labelled data set for the research.

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

## Appendix A. Optimal Cells Generated by ENAS for Breast Cancer Classification

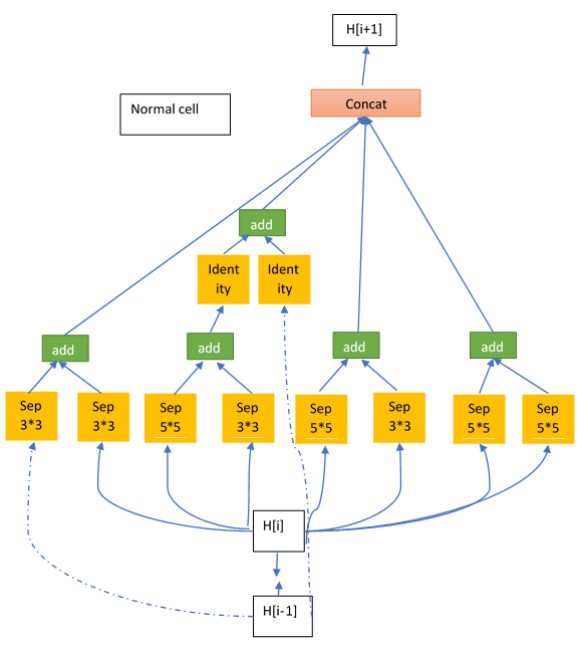

Figure 2: Normal Cell

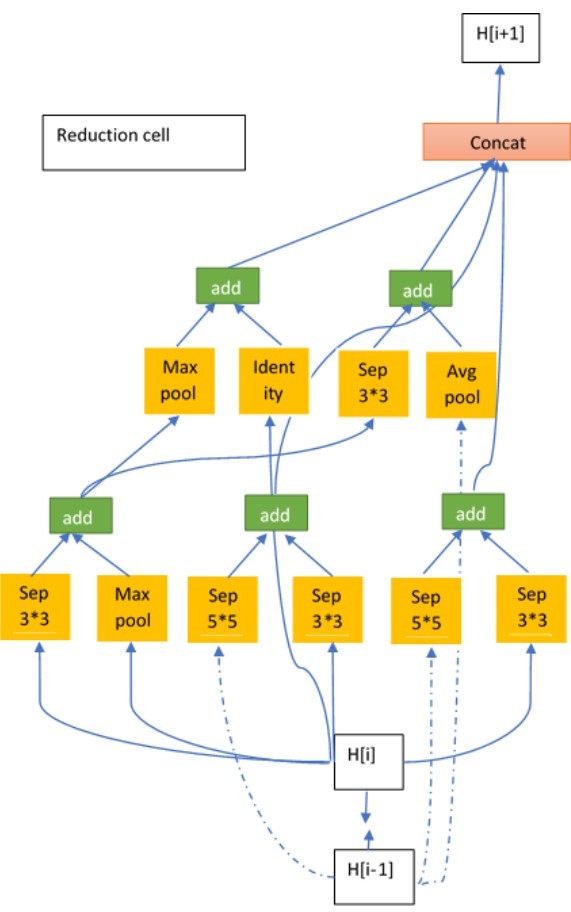

Figure 3: Reduction Cell

