# OpenReview forum: "An ENAS Based Approach for Constructing Deep Learning Models for Breast Cancer Recognition from Ultrasound Images"
_MIDL.io/2020/Conference — MIDL 2020_

### Official Review · AnonReviewer2 · 2020-02-20
**Apply ENAS to detect breast cancer using ultrasound**

**Rating:** 3
**Confidence:** 4

**Review:**

The authors implement use ENAS to classify benign and malignant breast lesions. It is an interesting approach that reaches an accuracy of 89.3%. The authors compare the ENAS results to AlexNet and CNN3 and show that their ENAS models give the highest performance. From the results it seems that the AlexNet just does not converge at all. The CNN3 network shows an accuracy of 78.1%, but it is unclear how these networks were trained.
Two small points:
(1) The authors mention that the input size is rescaled, but the original image size (in pixels and mm) is not mentioned and it is unclear if the acquired images were made with a preset, or that the sonographer was able to adjust zoom, gain etc.
(2) The authors augment the images using rotation of 90, 180 and 270 degrees, but this seems invalid since ultrasound images contain shadows which are always directed away from the transducer (so downwards).

---

### Official Review · AnonReviewer3 · 2020-03-11
**work on automatic optimization of neural network for breast cancer classification in ultrasound images: could be improved methodologically**

**Rating:** 2
**Confidence:** 5

**Review:**

pros:
- relatively large dataset of 524 US images (however, unclear from how many patients; potential bias)
- histologically verified classification
- automatic network optimization
- reasonable optimization time

cons:
- sloppy handling of references (examples see below)
- how do you ensure that the limited search space of ENAS does contain the optimal solution?
- how sensitive is the approach to scan parameters like frequency of ultrasound, directionality, Time gain compensation (depth compensation), reconstruction modes of US scanner, ...?
- data augmentation does not consider characteristics of ultrasound image formation
- boundaries of lesion only drawn by a single person
- only two classes are used; at least normal tissue would have been useful
- no sample images provided


comments:
I don't see how a paper named "Estimates of incidence and mortality of cervical cancer in 2018: a worldwide analysis" is a good reference for breast cancer. And indeed, scanning through the paper does not reveal much more than a few comparisons which backs your first sentence of the introduction. Please provide a more suitable reference.

Your sentence "However, no CNNs have been designed and optimized automatically (...)" is not fully correct. E.g. in Y. Weng, T. Zhou, Y. Li and X. Qiu, "NAS-Unet: Neural Architecture Search for Medical Image Segmentation," in IEEE Access, vol. 7, pp. 44247-44257, 2019. a UNET is used in combination of NAS for ultrasound nerve images. Since segmentation is a form of classification for individual voxels, your statement has to be corrected.

Type in caption of Figure 2: Redaction Cell -> Reduction Cell

---

### Official Review · AnonReviewer4 · 2020-03-12
**Using ENAS approach to search for the optimal CNN model design and testing the network on breast cancer data**

**Rating:** 2
**Confidence:** 3

**Review:**

This paper addresses one of the problems in deep learning, but the contribution is limited. The author uses an existing approach (ENAS) on a breast cancer dataset. So, it is not clear what is the author's contribution in terms of methodology.

The size of dataset is appropriate and the paper is well written, but the contribution is not very clear to me.

---

### Official Review · AnonReviewer1 · 2020-03-13
**Interesting experiment, but not clearly presented**

**Rating:** 3
**Confidence:** 4

**Review:**

This paper presents an interesting experiment that may be helpful for others too in finding better neural network parameters.
I'm not sure how many people at MIDL are familiar with ENAS. I'm not. Don't use acronym in the title. And explain better in the text what ENAS is.
In 2.1, you say you augmented the ultrasound image with 90, 180, etc. degrees rotation. Ultrasound is a directional imaging modality. Rotating beyond 15-20 degrees doesn't make any sense, and probably doesn't help in algorithm training either.
You also say that these augmentations create 7 extra images. Why not implement a data augmenter that manages training data and applies random rotations when the images are requested?
AlexNet is a bit old, and designed for two GPUs. I'm not sure why didn't you pick a more modern network.

---

### Meta-Review · Area_Chair1 · 2020-03-30
**MetaReview of Paper211 by AreaChair1**

**Rating:** 3

**Metareview:**

While this paper has received a mix of rates (2 weak accepts and 2 weak rejects), l tend towards rating this paper as weak accept. Nevertheless, I believe that authors need to address some important questions in the near future. As a summary, these are the main concerns of this work:
- Authors do not take into consideration the characteristics of ultrasound images. The motivation to augment this dataset (i.e., rotating 90, 180 and 270 degrees) seems invalid. I agree with the reviewers that this can not only not help, but also hurt the performance of the method.
-  Authors are encouraged to include higher performing models in their evaluation to strength the findings and results of this work.
- Furthermore, training on these models needs to be better detailed.

**Paper Type:**

validation/application paper

---

### Decision · Program_Chairs · 2020-04-11

Accept